# Effect of Different Durations of Solid-Phase Fermentation for Fireweed (*Chamerion angustifolium* (L.) Holub) Leaves on the Content of Polyphenols and Antioxidant Activity In Vitro

**DOI:** 10.3390/molecules25041011

**Published:** 2020-02-24

**Authors:** Marius Lasinskas, Elvyra Jariene, Nijole Vaitkeviciene, Ewelina Hallmann, Katarzyna Najman

**Affiliations:** 1Vytautas Magnus University. Agriculture Academy, Institute of Agriculture and Food Sciences, Donelaicio str. 58, 44248 Kaunas, Lithuania; elvyra.jariene@vdu.lt (E.J.); nijole.vaitkeviciene@vdu.lt (N.V.); 2Warsaw University of Life Sciences, Institute of Human Nutrition Sciences, Department of Functional and Organic Food, Nowoursynowska 15c, 02-776 Warsaw, Poland; ewelina_hallmann@sggw.pl (E.H.); katarzyna_najman@sggw.pl (K.N.)

**Keywords:** fermentation, fireweed, flavonoids, phenolic acids, tannin oenothein B

## Abstract

Fireweed has recently been recognized as a plant with high antioxidant potential and phenolic content. Its leaves can be fermented to prepare an infusion with ideal antioxidant activity. The aim of this study was to investigate and to determine the influence of solid-phase fermentation of different durations on the variation of polyphenols in the leaves of fireweed. Laboratory experiments were conducted in 2017–2018. The leaves of fireweed, naturally growing, were fermented for different periods of time: not fermented (control) and fermented for 24 and 48 h. The evaluation of polyphenols and antioxidant activity in leaves was performed using high- performance liquid chromatography (HPLC). Additionally, principal component analysis was used to characterize differences in bioactive compounds between fireweed samples fermented at different durations. Solid-phase fermented leaves were characterized by higher contents of oenothein B, quercetin and benzoic acid but had lower contents of quercetin-3-*O*-rutinoside, luteolin and chlorogenic and gallic acids. Antioxidant activity in short- (24 h) and long-term (48 h) fermentation (compared to control) gave the highest level of regression in 2017, but in 2018 the effect was observed only with short-term fermentation and control. In conclusion, solid-phase fermentation can be used to modulate biologically active compounds in fireweed leaves.

## 1. Introduction

Nowadays, food rich in antioxidants is very important. There are many plants well known to contain numerous biologically active substances. These plants include tea (*Camellia sinensis* (L.) Kuntze) and coffee (*Coffea arabica*). Food and beverages with high antioxidant potential are produced from these plants [1,2]. On the other hand, new plants with such properties are constantly being sought. One of the plants that can help to meet health and nutrition needs is the narrow-leaved fireweed (*Chamerion angustifolium* (L.) Holub), which is one of the best known medicinal plants in Lithuania. The biological activity of a plant depends on the place where it grows, the meteorological conditions during vegetation, the growth phase and other factors. In Lithuania, fireweed grows in a variety of soils, but most often in damaged areas of the earth (cut or burned forests, soils, highways and railroad tracks). It can grow in light forests, but not in full shade. In other countries it is found in both marine and continental climate zones with short hot summers and long cold winters. Annual rainfall in these habitats can range from 330 to 3420 mm, with temperatures ranging from −50 to 30 °C and altitudes from 0 to 4700 m above sea level [3]. Fireweed leaves are used in traditional and folk medicine and have various pharmacological effects: antioxidant, anti-inflammatory, analgesic, anticancer and others [4,5,6]. There are a lot of data on the beneficial effects of fireweed leaf tea in the treatment of anemia, headaches, infections and colds; the leaves are also useful in the treatment of various gastrointestinal disorders and prostate and urinary problems [7]. It is very important to find a way to increase the bioavailability of polyphenolic compounds from plants [8]. It seems that not only the high content of bioactive compounds in plants, but also the level of their availability in infusion is essential [9]. In this light, it is very important to examine the chemical composition of the leaves of the fireweed and the method of their fermentation. One of the ways to modulate biologically active compounds and their bioavailability in fireweed leaves is to use solid-phase fermentation [10].

Fireweeds are an ideal source of polyphenolic compounds. In their leaves, many flavonoids and phenolic acids have been identified: quercetin-3-*O*-rutinoside, myricetin, luteolin, quercetin, quercetin-3-*O*-glucoside, kaempferol, gallic and ellagic acids, tannin oenothein B and many others [11,12,13].

However, there are not enough studies on the influence of different solid-phase fermentation durations on the chemical composition of fireweed leaves. The aim of this study was to investigate the influence of solid-phase fermentation of different durations on the content of flavonoids, tannin oenothein B and phenolic acids in fireweed leaves. Based on the results of this study, solid-phase fermentation optimization possibilities can be recommended to allow manufacturers to produce high-quality products from fireweed leaves.

## 2. Results

### 2.1. Content of Bioactive Compounds

The effect of fermentation process changed the level of polyphenols in fireweed leaves in both years of the experiment (Table 1).

In the second year of research, when meteorological conditions during vegetation were less favorable due to lack of moisture for the development of fireweed, possibly due to stress, the plant accumulated higher amount of all bioactive compounds (control). It is likely that this may have been one of the factors explaining the differences in results between years. We observed the effects of 24 and 48 h fermentation processes on all polyphenolic groups in 2017. It is worth noting that long-term fermentation (48 h) decreases the level of all phenolic compounds from different chemical phenolic groups compared with short-term (24 h) fermentation. In 2018, it seems that long term fermentation was more effective in case of total polyphenols content (Table 1). A similar situation was observed with individual phenolic compounds especially in 2017. Only in 2018 did we observe a significantly higher level of bioactive compounds in nonfermented fireweed leaves. Short-term fermentation significantly increased the level of quercetin in 2017 and 2018 (Table 2 and Table 3).

The level of kaempferol was significantly higher after long-term fermentation (48 h) only in 2017 (Table 2 and Table 3).

### 2.2. Antioxidant Activity in Fireweed Leaves

We observed that, in 2017, nonfermented fireweed was characterized by a lower antioxidant activity compared to short (24 h) and long (48 h) fermentation processes. In 2018, control samples were characterized by a higher antioxidant activity compared to experimental combinations. It is worth noting that only the short fermentation time gave positive results in antioxidant activity of fireweeds in vitro (Figure 1).

On the other hand, we also noticed that there was strong regression between antioxidant activity (in mg 100 g^−1^ DW Trolox equivalents) and the total content of polyphenols in fireweed leaves depending on the time of fermentation used. In 2017, both methods of fermentation (compared to control) gave the highest level of regression. In 2018 the effect was observed only with the short time of fermentation and control (Table 4).

### 2.3. Distribution of Polyphenols in Different Fireweed Samples

Principal component analysis (PCA) was performed to establish the relationship among fireweed samples fermented for different durations (not fermented, fermented 24 h and fermented 48 h) and polyphenols, regarding the levels of these compounds in the samples in 2017 and 2018. 

PCA was carried out on the 3 fireweed samples and 15 parameters shown in Table 1, Table 2 and Table 3. As depicted in Figure 2A, the first two PCA axes (PC1 and PC2) characterized 100% of the total variance in polyphenols of fireweed samples from the 2017 season. Quercetin-3-*O*-rutinoside, luteolin, chlorogenic acid and gallic acid were positively related with PC1 (80.82% of variance), while total polyphenols, total flavonoids, myricetin, total phenolic acids, oenothein B, quercetin and p-coumaric, ellagic and benzoic acids were negatively related with PC1 (Figure 2A). PC2 positively correlated with levels of quercetin-3-*O*-glucoside and kaempferol (19.18% of variance) (Figure 2A).

As shown in Figure 2B, the control (not fermented) sample positively correlated on PC1 was described by high levels of quercetin-3-*O*-rutinoside, luteolin, chlorogenic acid and gallic acid. Sample fermented 24 h had negative values of PC1. Consequently, this sample was characterized as containing high levels of total polyphenols, total flavonoids, myricetin, total phenolic acids, oenothein B, quercetin and p-coumaric, ellagic and benzoic acids. Fireweed sample fermented 48 h had positive PC2 values. Therefore, this sample was defined by high levels of quercetin-3-*O*-glucoside and kaempferol.

The PC1 and PC2 of the 2018 growing season described 100% of the total variance: PC1 represented 77.27% and the PC2 represented 22.73%. Total flavonoids, oenothein B and quercetin were positively associated with PC1, whereas total polyphenols, total phenolic acids, myricetin, luteolin, kaempferol and gallic, chlorogenic, p-coumaric and ellagic acids were negatively related with PC1 (Figure 3A). Quercetin-3-*O*-rutinoside, quercetin-3-*O*-glucoside and benzoic acid were positively associated with PC2 (Figure 3A).

The projection of fireweed samples fermented for different durations in 2018 is located in Figure 3B. Sample fermented 24 h positively correlated on PC1 was described by high levels of total flavonoids, oenothein B and quercetin, while control (not fermented) fireweed sample negatively correlated on the PC1 was defined by high levels of total polyphenols, total phenolic acids, myricetin, luteolin, kaempferol and gallic, chlorogenic, *p*-coumaric and ellagic acids. For the 2018 season, the PC2 clarified 22.73% of the total variance. Fireweed sample fermented 48 h positively correlated on PC2 was characterized by high levels of quercetin-3-*O*-rutinoside, quercetin-3-*O*-glucoside and benzoic acid.

## 3. Discussion

Phenolic compounds in fireweed are secondary metabolites used by the plant to reduce the effects of stress. In particular, the synthesis of these compounds is intensified in response to climatic, moisture or other stress-causing effects on the plant. The meteorological conditions in the experimental years varied widely, and the different results for certain materials could be explained by the fact that 2017 was rainy and cool, but 2018 was dry and warm compared to the standard climate normal. The year 2018 was too dry for the fireweed development and thus could be one of the main factors for observation of higher amounts of polyphenols than in 2017.

Microbial metabolism and enzymes produced during the process of solid-phase fermentation have an important influence on the composition of fermented products by disintegrating macromolecular components (lipids, polysaccharides, proteins) into lower molecular weight compounds (free fatty acids, dextrins, sugars, peptides, amino acids) and secondary products of metabolism (organic acids, esters, aldehydes, vitamins, etc.) [14]. 

There is no united information on how fermentation process can influence the level of bioactive compounds. In the presented experiment we observed that fermentation process increased the level of phenolic compounds, especially flavonoids. Kosman et al. 2013 confirmed that phenomenon [15]. However, other results were obtained by Kauppinen and Galambosi (2016), showing the opposite [16]. After short-term (24 h) fermentation the level of the total polyphenols decreased from 4310 to 3956 mg 100 g^−1^ dry weight (DW) in fireweed leaves in 2018. The fermentation process changed the quantity and quality of flavonoids. As reported by Hallmann et al. (2017) [17], significantly lower total flavonoid content was found in fermented white cabbage juice (sauerkraut) compared to fresh juice. 

The flavonoids (quercetin) found in glucoside form increased significantly after fermentation. Free aglycone forms appeared (quercetin) (Table 2 and Table 3). This is probably due to the breakdown of glycoside bonds by fermentation bacteria [18].

Quercetin is the aglycone form of a number of other flavonoid glycosides, such as rutin (also named quercetin-3-*O*-rutinoside) and quercitrin. As we observed, these flavonoids are closely related to each other in structural and biochemical terms, which can explain the results among flavonoid quantities (Figure 2 and Figure 3).

Myricetin quantity decreased by 42.60% after 24 h and 39.16% after 48 h fermentation in 2018 compared with control; in 2017, no significant differences among variants were found. This could be explained by activity of different enzymes and microorganisms in higher amounts of biologically active substances (flavonoids) as weather conditions during fireweed vegetation varied: 2017 was rainy and cool, but 2018 was dry and warm. So in this case, flavonoids (myricetin) in control in 2018 were produced more (18.31 mg 100 g^−1^ DW) than in 2017 (8.91 mg 100 g^−1^ DW) and thus could have influenced the fermentation process. Furthermore, myricetin is structurally similar to luteolin and quercetin. Myricetin can alternatively be produced directly from kaempferol, which is another flavonol [19]. The level of kaempferol was significantly higher (30.81%) after long-term fermentation (48 h) only in 2017.

Gallic acid quantities after fermentation decreased in both years: after 24 h 58.59% in 2017 and 47.03% in 2018; after 48 h fermentation decreased 52.83% (2017) and 46.11% (2018) compared with control. Gallic acid is found both in free form and as part of hydrolysable tannins. Gallic acid groups are related to dimers such as ellagic acid. Hydrolysable tannins are broken down during hydrolysis into gallic acid and glucose or ellagic acid and glucose, called gallotannins and ellagitannins [20]. 

Ellagic acid is the most abundant phenolic acid present in the leaves of the fireweed. Plants produce ellagic acid by hydrolyzing tannins such as ellagitannin and geraniol [21]. In this case, solid-phase fermentation significantly reduced the levels of ellagic acid both after 24 h (21.67%) and 48 h (16.25%) fermentation in 2018 compared with control (Table 3). The decrease in the ellagic acid can be explained by the fact that as microbial metabolism and enzyme activity intensifies during fermentation, part of the ellagic acid is broken down to the final acid.

Oenothein B is a unique macrocyclic ellagitannin dimer whose quantity is related to ellagic and gallic acids quantities. In both years, after all fermentations the level of oenothein B increased, especially after 24 h of fermentation (Table 2 and Table 3). This can be explained by the decrease of gallic acid during solid-phase fermentation, as microbial metabolism of ellagic acid and galotanins intensifies and the enzyme activity breaks them up into lower molecular weight compounds [22]. 

Benzoic acid is in many plants and mediates the biosynthesis of many secondary metabolites. Benzamide and benzonitrile can be hydrolyzed to benzoic acid or its conjugated base under acidic or alkaline conditions [23]. The content of benzoic acid in fireweed was significantly increased after 24 h of solid-phase fermentation, and 48 h solid-phase fermentation significantly reduced the amount of this phenolic acid in 2017 (Table 2). In 2018, the tendencies were opposite (Table 3). Significant changes in benzoic acid could be explained by the fact that anaerobic bacterial metabolism and active enzyme activity initially (after 24 h) induced the formation of benzoic acid from macromolecular compounds, but later (after 48 h) the benzoic acid was incorporated into other compounds (e.g., gallic acid increased).

There is a close relationship between chlorogenic and p-coumaric acids, since cinnamic acid participates in the metabolism of both [24]. Chlorogenic acid levels decreased significantly after 24 and 48 h solid-phase fermentations compared to control in both years (Table 2 and Table 3). *p*-Coumaric acid after 24 h and 48 h solid-phase fermentation volume increased significantly only in 2017 (Table 2). 

Other scientists [25] suggest that the final quality of a solid-phase fermentation process can be influenced not only by the duration of the fermentation, but also by other factors, such as certain microorganisms: bacteria, yeast, fungi and other. 

Polyphenol concentration in fireweed leaves reflects their antioxidant activity [26]. These results was confirmed in our study. At the same time, we noticed changes in the effectiveness of the fermentation process between years, which could be connected with different weather conditions effects on fireweed vegetation. 

## 4. Materials and Methods

### 4.1. Chemicals

ABTS (2,2′-azino-bis(3-ethylbenzothiazoline-6-sulfonic acid) diammonium salt (Sigma-Aldrich, Poland), acetonitrile HPLC purity (Sigma-Aldrich, Poznan, Poland), deionized water (Sigma-Aldrich, Poland), methanol HPLC purity (Merck, Warszawa, Poland), ortho-phosphoric acid 99.9% (Chempur, Piekary Śląskie, Poland), phenolics standards (purity 99.5%–99.9%) of gallic, chlorogenic, *p*-coumaric benzoic, ellagic, oenothein B, quercetin-3-*O*-rutinoside, myricetin, luteolin, quercetin, quercetin-3-*O*-glucoside and kaempferol (Sigma-Aldrich, Poland) and phosphate-buffered saline (Merck, Warszawa, Poland) were used.

### 4.2. Plants Origin

The leaves of fireweed (*Chamerion angustifolium* (L.) Holub), naturally growing (in Ausrine Semiene biodynamic farm, in Svencionys district, Dauksiai village, Lithuania,) were investigated in 2017 and 2018. During the experiment, the meteorological conditions in the Eastern Lithuania region, Svencionys district, were evaluated. Changes in meteorological conditions had an important influence on the leaf size of the fireweed plant, its chemical composition and the amount of biologically active compounds. During the vegetation period of 2017, rainy and cooler weather prevailed (rainfall sum in May–September period was 313 mm; air temperature was 15.2 °C); in 2018 there was drier and warmer weather (rainfall sum in May–September period was 273 mm; air temperature was 17.5 °C), compared to the standard climate normal (SCN) (Table 5). During the drier period in 2018, a stressful situation developed and was apparently one of the factors influencing the biochemical composition in fireweed leaves.

### 4.3. Plants Material Preparation and Solid-Phase Fermentation Process

The raw material was randomly collected from different places of the site in July at the beginning of mass flowering. The combined leaves sample was 5.4 kg. For laboratory experimentation, it was divided into three parts: 1.8 kg (dried, unfermented) for control and 3.6 kg for solid-phase fermentation lasting 24 and 48 h. Control (0 h) was not fermented but stored for the intended time. 

In the solid-phase fermentation, fresh fireweed leaves were cut with special plastic knives and the resulting raw material was divided into 0.600 kg. The mass ready for fermentation was rigidly pressed into glass containers and covered with a lid. The fermentation process took place at 30 °C in a dark chamber for 24 and 48 h. Each experimental variation was run in three replications. After fermentation, the raw material was dried at 40 °C and stored in a dark, dry, cool and ventilated room. All chemical tests were performed in triplicate.

### 4.4. Polyphenol Separation and Identification

Polyphenol compounds were measured by an high-performance liquid chromatography (HPLC) method that was described previously in detail by Ponder and Hallmann (2019) [27]. A total of 100 mg of powdered dried plant tissue was mixed with 5 mL of 80% methanol and shaken on a Micro-Shaker 326 M (Marki, Poland). Next, all samples were extracted in an ultrasonic bath (10 min, 30 °C, 5.5 kHz). Fifteen minutes after extraction, the samples were centrifuged (10 min, 3780× *g*, 5 °C). The supernatant was carefully collected in a clean plastic tube and centrifuged again (5 min, 31,180× *g*, 0 °C). A total of 850 μL of supernatant was transferred to an HPLC vial and analyzed. For polyphenol compound separation and identification, a Synergi Fusion-RP 80i Phenomenex column (250 × 4.60 mm) was used. The analysis was carried out with the use of Shimadzu equipment (USA Manufacturing Inc, Lebanon, IN, USA: two pumps LC-20AD, controller CBM-20A, column oven SIL-20AC, spectrometer UV/Vis SPD-20 AV). The phenolic compounds were separated under gradient conditions with a flow rate of 1 mL min^−1^. Two gradient phases were used: 10% (*v*/*v*) acetonitrile and ultrapure water (phase A) and 55% (*v*/*v*) acetonitrile and ultrapure water (phase B). The phases were acidified by orthophosphoric acid (pH 3.0). The total time of the analysis was 38 min. The phase-time program was as follows: 1.00–22.99 min, 95% phase A and 5% phase B; 23.00–27.99 min, 50% phase A and 50% phase B; 28.00–28.99 min, 80% phase A and 20% phase B; and 29.00–38.00 min, 95% phase A and 5% phase B. The wavelengths were 250 nm for flavonols and 370 nm for phenolic acids. The phenolic compounds were identified by using 99.9% pure standards (Sigma-Aldrich, Poland) and the analysis times for the standards are shown in Figure 4 and Figure 5.

### 4.5. Antioxidant Activity

The procedure of antioxidant activity measurement was described in detail previously by Srednicka-Tober et al. [28]. Briefly, 250 mg of the freeze-dried plant powder was weighed into a plastic tube, and 25 mL of distilled water was added. Samples were mixed on the vortex (Labo Plus, Warsaw, Poland) for 1 min. Next, the samples were incubated in a shaker incubator (IKA, Staufen im Breisgau, Germany) for 1 h (temperature 30 °C). After incubation, the sample was shaken again and then centrifuged (Centrifuge, MPW-380 R, Warsaw, Poland) at 5 °C and 14,560× *g* for 15 min. In the next phase, the supernatant was collected for determinations. In laboratory glass tubes, examined plant extract solutions were measured with a predetermined dilution scheme (0.5–1.5 mL) and then added to 3.0 mL of ABTS·+ cationic solution in PBS (phosphate-buffered saline). After 6 min, sample absorbances were taken (21 °C, wavelength λ = 734 nm) using a spectrophotometer (Helios γ, Thermo Scientific, Warsaw, Poland). The obtained measurements were calculated using a special formula including the dilution factor. The final results were express as mmol of TE (Trolox equivalents/100 g DW).

### 4.6. Statistical and Multivariate Analysis

All data were statistically processed using a two-way analysis of variance (ANOVA) method from the STATISTICA software package (Statistica 10; StatSoft, Inc., Tulsa, OK, USA). Data were expressed as the mean value ± standard error. Number of repetitions *n* = 3, number of replications each combination *n* = 2. The statistical significance of differences between the means was estimated by Fisher′s LSD test (*p* < 0.05). Principal components analysis (PCA) was carried out using XLSTAT Software (XLSTAT, 2018, New York, NY, USA) to categorize the fireweed samples fermented for different durations based on their bioactive compounds.

## 5. Conclusions

The experiment showed that the leaves of fireweed fermented for different durations differ significantly in the content of polyphenolic compounds. Solid-phase fermentation increased the levels of oenothein B, quercetin and benzoic acid, but had lower contents of quercetin-3-*O*-rutinoside, luteolin and chlorogenic and gallic acids. The results of this study were confirmed by PCA. Despite year-to-year effects, a clear clustering related to the effects of different durations of fermentation was shown. The strong regression between antioxidant activity and the total content of polyphenols in fireweed leaves depended on the meteorological conditions during vegetation period and time of fermentation: in rainy and cool 2017, both methods of fermentation (compared to control) gave the highest level of regression; in dry and warm 2018, the effect was observed only with shorter time of fermentation and the control. In conclusion, solid-phase fermentation can be used to modulate biologically active compounds in fireweed leaves, which can be used to prepare health-promoting and disease-preventive products. It has to be considered that the proposed method will give the assumed results with fireweed (*Chamerion angustifolium* (L.) Holub) leaves or plants with such composition.

## Figures and Tables

**Figure 1 molecules-25-01011-f001:**
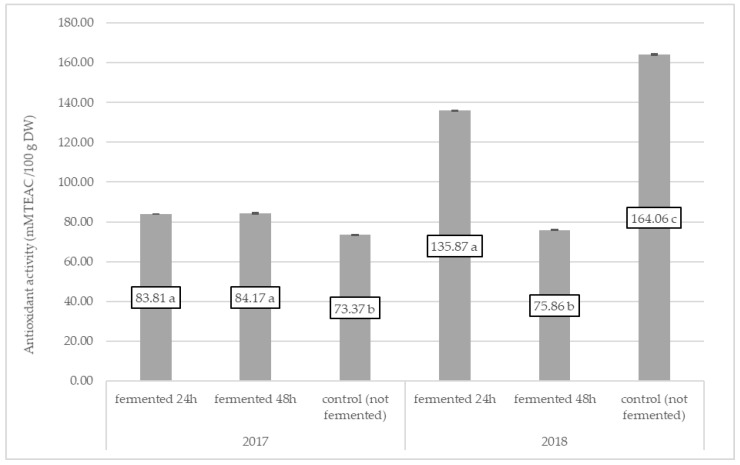
Antioxidant activity of fireweed leaves in 2017 (*p* < 0.0001) and 2018 (*p* < 0.0001) Means followed by the same letter are not significantly different (*p* < 0.05), *n* = 3.

**Figure 2 molecules-25-01011-f002:**
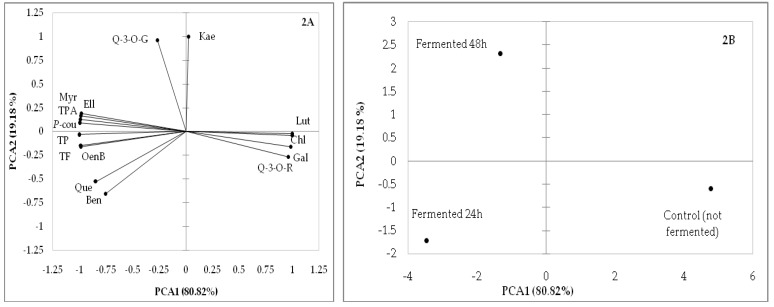
PCA results for the 2017 growing season: (**2A**) factor loadings for polyphenols (total polyphenols (TP), total flavonoids (TF), quercetin-3-*O*-rutinoside (Q-3-*O*-R), myricetin (Myr), luteolin (Lut), quercetin (Que), quercetin-3-*O*-glucoside (Q-3-*O*-G), kaempferol (Kae), total phenolic acids (TPA), gallic (Gal), chlorogenic (Chl), p-coumaric (P-cou), ellagic (Ell), benzoic (Ben), oenothein B (OenB)) and (**2B**) projection of fireweed samples fermented for different durations (control (not fermented), fermented 24 h and fermented 48 h).

**Figure 3 molecules-25-01011-f003:**
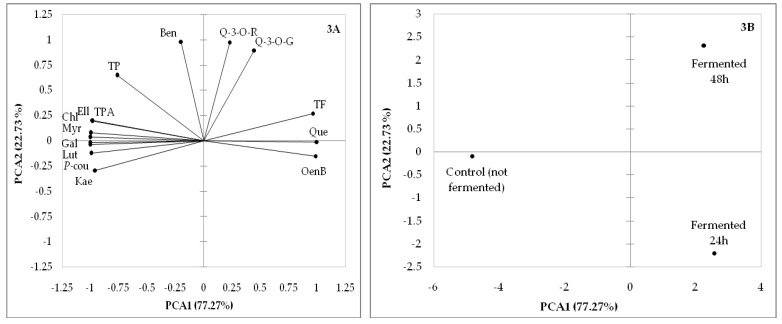
PCA results for the 2018 growing season: (**3A**) factor loadings for polyphenols (total polyphenols (TP), total flavonoids (TF), quercetin-3-O-rutinoside (Q-3-*O*-R), myricetin (Myr), luteolin (Lut), quercetin (Que), quercetin-3-*O*-glucoside (Q-3-*O*-G), kaempferol (Kae), total phenolic acids (TPA), gallic (Gal), chlorogenic (Chl), *p*-coumaric (*P*-cou), ellagic (Ell), benzoic (Ben), oenothein B (OenB)) and (**3B**) projection of fireweed samples fermented for different durations (control (not fermented), fermented 24 h and fermented 48 h).

**Figure 4 molecules-25-01011-f004:**
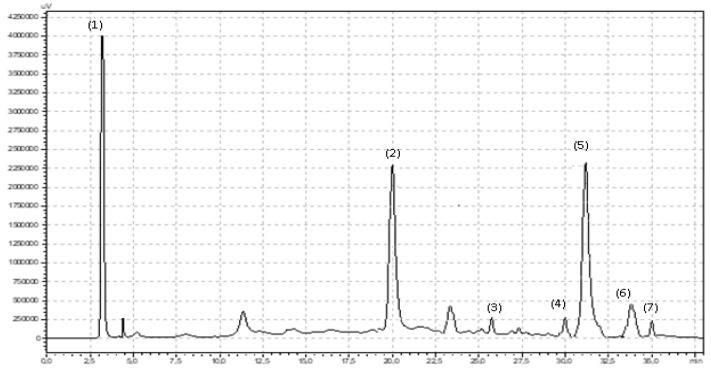
Chromatogram showing retention times for flavonoids in fireweed leaves: (**1**) oenothein B, (**2**) quercetin-3-O-rutinoside, (**3**) myricetin, (**4**) luteolin, (**5**) quercetin, (**6**) quercetin-3-O-glucoside, (**7**) kaempferol.

**Figure 5 molecules-25-01011-f005:**
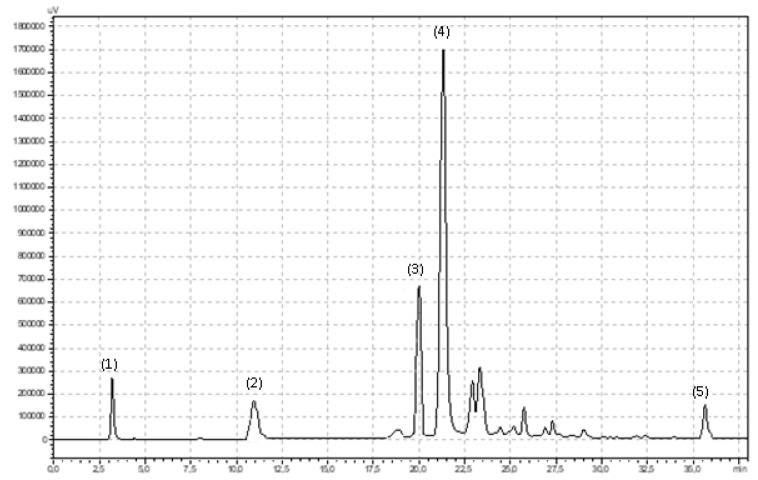
Chromatogram showing retention times for phenolic acids in fireweed leaves: (**1**) gallic, (**2**) chlorogenic, (**3**) p-coumaric (**4**) benzoic, (**5**) ellagic.

**Table 1 molecules-25-01011-t001:** The content of the total bioactive compounds (mg 100 g^−1^ dry weight (DW)) from different polyphenols groups in fireweed leaves. Mean value ± standard error, *n* = 3.

Bioactive Group/Fermentation Method	2017			
	Fermented 24 h	Fermented 48 h	Control (Not Fermented)	*p*-Value
Total polyphenols	3506.80 ± 14.17 a	3024.96 ± 20.88 b	1843.01 ± 38.22 c	<0.00001
Total phenolic acids	1802.27 ± 33.70 a ^1^	1712.63 ± 16.22 a	1131.46 ± 21.81 b	<0.00001
Total flavonoids	1704.52 ± 23.41 a	1312.33 ± 11.94 b	711.55 ± 21.06 c	<0.00001
**Bioactive Group/** **Fermentation Method**	**2018**			
	**Fermented 24 h**	**Fermented 48 h**	**Control (Not Fermented)**	***p*-Value**
Total polyphenols	3956.00 ± 9.80 b	4200.50 ± 28.05 a	4309.77 ± 13.80 a	<0.00001
Total phenolic acids	2135.62 ± 11.75 b	2289.14 ± 27.72 b	2772.18 ± 14.64 a	<0.00001
Total flavonoids	1820.38 ± 5.01 a	1911.37 ± 8.73 a	1537.60 ± 14.59 b	<0.00001

^1^ Means in rows followed by the same letter are not significantly different at the 5% level of probability (*p* < 0.05).

**Table 2 molecules-25-01011-t002:** The content of individual bioactive compounds (mg 100 g^−1^ DW) in fireweed leaves in 2017. Mean value ± standard error, *n* = 3.

Compound/Fermentation Method	Fermented 24 h	Fermented 48 h	Control (Not Fermented)	*p*-Value
gallic	4.17 ± 0.22 b ^1^	4.75 ± 0.12 b	10.07 ± 0.07 a	<0.00001
chlorogenic	4.78 ± 0.08 c	5.42 ± 0.13 b	7.70 ± 0.09 a	0.02000
*p*-coumaric	114.38 ± 2.37 a	104.88 ± 1.55 a	58.63 ± 1.27 b	<0.00001
ellagic	1649.14 ± 35.71 a	1592.29 ± 16.41 a	1052.06 ± 20.96 b	<0.00001
benzoic	29.81 ± 0.55 a	5.30 ± 0.13 b	3.00 ± 0.10 c	<0.00001
oenothein B	1624.81 ± 23.0 a	1214.43 ± 11.75 b	611.72 ± 20.54 c	<0.00001
quercetin-3-*O*-rutinoside	27.20 ± 0.16 b	26.59 ± 0.41 b	53.14 ± 0.48 a	<0.0001
myricetin	11.26 ± 0.22 b	11.09 ± 0.12 b	8.91 ± 0.33 a	N.S. ^2^
luteolin	3.59 ± 0.04 b	3.80 ± 0.02 b	4.49 ± 0.02 a	<0.00001
quercetin	10.57 ± 0.09 a	6.94 ± 0.10 b	5.76 ± 0.09 b	<0.00001
quercetin-3-*O*-glucoside	24.23 ± 0.44 b	45.13 ± 0.66 a	24.23 ± 0.71 b	<0.00001
kaempferol	2.87 ± 0.04 b	4.33 ± 0.04 a	3.31 ± 0.02 b	<0.00001

^1^ Means in rows followed by the same letter are not significantly different at the 5% level of probability (*p* < 0.05); ^2^ N.S., not significant.

**Table 3 molecules-25-01011-t003:** The content of individual bioactive compounds (mg 100 g^−1^ DW) in fireweed leaves in 2018. Mean value ± standard error, *n* = 3.

Compound/Fermentation Method	Fermented 24 h	Fermented 48 h	Control (Not Fermented)	*p*-Value
gallic	6.33 ± 0.12 b ^1^	6.44 ± 0.10 b	11.95 ± 0.09 a	<0.00001
chlorogenic	8.10 ± 1.22 b	9.54 ± 0.13 b	19.48 ± 0.07 a	<0.00001
*p*-coumaric	87.01 ± 2.18 b	81.91 ± 1.32 b	134.67 ± 6.89 a	<0.00001
ellagic	2029.71 ± 11.95 b	2169.42 ± 28.75 b	2590.48 ± 17.05 a	<0.00001
benzoic	4.47 ± 0.15 c	21.82 ± 0.17 a	15.59 ± 0.60 b	<0.00001
oenothein B	1753.65 ± 4.82 a	1736.47 ± 7.24 a	1442.22 ± 13.55 b	<0.00001
quercetin-3-*O*-rutinoside	25.85 ± 0.72 c	108.50 ± 1.60 a	47.28 ± 0.70 b	<0.00001
myricetin	10.51 ± 0.39 b	11.14 ± 0.08 b	18.31 ± 0.11 a	<0.00001
luteolin	2.37 ± 0.01 b	2.40 ± 0.01 b	6.33 ± 0.16 a	<0.00001
quercetin	8.75 ± 0.04 a	7.48 ± 0.04 b	2.39 ± 0.20 c	<0.00001
quercetin-3-*O*-glucoside	16.54 ± 0.42 b	43.10 ± 0.63 a	17.20 ± 0.34 b	<0.00001
kaempferol	2.70 ± 0.02 b	2.27 ± 0.02 b	3.87 ± 0.03 a	<0.00001

^1^ Means in rows followed by the same letter are not significantly different at the 5% level of probability (*p* < 0.05).

**Table 4 molecules-25-01011-t004:** The simple Pearson’s regression between antioxidant activity and total polyphenols in fireweed leaves for two experimental years 2017 and 2018.

2017	R^2^	Correlation Coefficient (in %)	*p*-Value
Fermented 24 h	0.91	95.37	0.0032
Fermented 48 h	0.84	91.71	0.0100
Control (not fermented)	0.71	84.38	0.0340
**2018**			
Fermented 24 h	0.72	84.97	0.0320
Fermented 48 h	0.59	77.28	N.S. ^1^
Control (not fermented)	0.90	95.09	0.0036

^1^ N.S., not significant.

**Table 5 molecules-25-01011-t005:** Weather conditions for fireweed growing season 2017 and 2018 (Vilnius weather station).

Years	Months					Average, °C
	May	June	July	August	September	
Air temperature, °C						
2017	12.9	15.4	16.8	17.5	13.4	15.2
2018	17.0	17.3	19.5	19.1	14.6	17.5
The standard climate normal ^1^	12.9	15.7	18.1	17.2	12.3	15.2
Rainfall, mm						Sum, mm
2017	11	80	80	55	87	313
2018	27	16	108	65	57	273
The standard climate normal	57	73	89	75	66	360
Sunshine, h						Sum, h
2017	285	225	215	230	145	1100
2018	365	284	209	278	206	1342
The standard climate normal	252	246	260	237	154	1149

^1^ Standard climate normal (SCN) is a 30 year average from 1981 to 2010.

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
