# Peer review of "Effect of Different Durations of Solid-Phase Fermentation for Fireweed (Chamerion angustifolium (L.) Holub) Leaves on the Content of Polyphenols and Antioxidant Activity In Vitro"

_molecules, 2020, doi:10.3390/molecules25041011_

Round 1

Reviewer 1 Report

The manuscript: “Optimization of solid-phase fermentation for fireweed (Chamerion angustifolium (L.) Holub), its effect on the content of polyphenols and antioxidant activity in vitro” describes the influence of solid-phase fermentation in different duration on the variation of polyphenols in leaves of fireweed. I find the manuscript presents interesting results, however, in my opinion,  the major revision has to be incorporated before publication. The shown data could rather be described in the short form of article. To improve the quality of work, the authors should decide to carry out additional tests, e.g. a greater variety of tested extracts obtained under different conditions or to extend biological tests that could indicate that the new extracts have a different biological potential from control. I have many comments which should help to improve the publication.

The title is not entirely accurate. Optimization suggests extensive work on choosing the conditions of the experiment while two-time points were selected, without thinking about additional parameters.

The Abstract should change. The background of the experiment was described too superficially, information on research methods is too general, similar the results cited.

Keywords should be different from those in the title to facilitate the dissemination of the work.

The Introduction should also be refined. The information is arranged somewhat chaotically. There is also a statement that fireweed is "the best known in ... worldwide", which I think is exaggerated. Besides, there could be some information about the occurrence or availability of raw material even in the country of the authors.

In the Results section, there is no information about the antioxidant activity, how it was tested, how many methods, what activity the raw material had and how it differed depending on the tested sample. I did not find such information in the discussion, which is mainly a presentation of the results of the contents of individual compounds. The authors also highlight various climatic conditions in which the raw material grew, however, there is no more detailed description, e.g. the amount of precipitation in a given period or temperature.

The methodology lacks a few sentences, even about the type of methods for determining antioxidant activity. I also didn't notice the way the content results were expressed. The methodology also lacks the “Chemicals” section to characterize the reagents used.

There is also a question whether in Discussion/Conclusion there should not be a sentence about limitations such as obtaining plants with such composition that the proposed method will give the assumed results.

Reviewer 2 Report

The presented work describes important problems. However, any analytical technique used must be validated. For this reason, the authors should provide the accuracy, precision and repeatability of the HPLC technique used in the context of the individual bioactive compounds determined. Authors must also provide the appropriate reference on page 2, line 59.

Round 2

Reviewer 1 Report

The authors have edited the manuscript and made the recommended changes. I think it is ready for publication.

Reviewer 2 Report

A revised manuscript may be published.